# Protocol for the development of a core outcome set for respectful maternal and newborn care in a low-middle income setting

Farai Marenga[1], Kushupika Dube[1], Unice Goshomi[1], Tina Lavender[2], Carol Bedwell[2], Jamie J. Kirkham [ID][3]*

1 Women's University in Africa, Harare, Zimbabwe, 2 Department of International Public Health, Liverpool School of Tropical Medicine, Liverpool, United Kingdom, 3 Centre for Biostatistics, The University of Manchester, Manchester Academic Health Science Centre, Manchester, United Kingdom

* jamie.kirkham@manchester.ac.uk

## Abstract

### Introduction

Disrespect and abuse have been seen as a real hindrance to achieving universal coverage for skilled delivery. Improving respectful maternal and newborn care and quality of care around the time of birth has been identified as a key strategy in low- and middle-income countries for reducing the rates of stillbirths and maternal and newborn mortality and morbidity rates. Currently, there is no core outcome set on respectful maternal and newborn care, resulting in reporting of various study outcomes from different studies which hinders the improvement of maternal and neonatal health.

### Objective

To develop a core outcome set for respectful maternal and newborn care that can be used in research studies and clinical practice in low-middle income countries.

### Methods/design

An exploratory sequential mixed methods evidence synthesis design will be adopted for the study. This design will enable the utilisation of the core outcome set development methodology in three stages. First, a systematic review and secondary analysis of qualitative interviews of women who utilise maternal care services will be undertaken in order to generate a list of outcomes. This will be followed by a two-round online Delphi study with multiple stakeholder groups which include women and their partners, women representative groups, parents, health workers and researchers. Each person will score the outcomes in terms of the defined criteria. Lastly, the results of the Delphi will be summarised and discussed at a virtual consensus meeting with representation from all stakeholder groups where the final core outcome set will be decided.

**Data availability statement:** No datasets were generated or analysed during the current study as it is a protocol. All relevant data from this study will be made available upon study completion.

**Funding:** This research is funded by the National Institute for Health Care Research (NIHR) (NIHR 132027), a major funder of global health research and training, using UK aid from the UK Government to support global health research. The views expressed in this publication are those of the author(s) and not necessarily those of the NIHR or the UK Department of Health and Social Care.

**Competing interests:** JJK is a member of the COMET management group. All other authors have no competing interests. This does not alter our adherence to PLOS ONE policies on sharing data and materials.

## Discussion

The core outcome set will predominantly be developed for use in a low-middle income country setting to measure and improve the quality of respectful maternal and newborn services.

## Introduction

The World Health Organisation (WHO) and United Nations International Children's Education Fund (UNICEF), reports that there are approximately 7,000 newborns and 800 women that die daily due to pregnancy and birth related complications, and most of these deaths occur in sub-Saharan Africa [1].

Institutional births by skilled health professionals have been identified as a factor that can greatly reduce the maternal and newborn mortality rates [2]. However, Low-Middle Income Countries (LMICs) access to quality health services is not always guaranteed and even when the services are available and accessible the quality may be compromised by disrespect, abuse and mistreatment during childbirth [3]. Disrespect and abuse have been seen as a hindrance to achieving universal coverage for skilled care at birth. Therefore, improving respectful maternal and newborn care (RMNC) and quality of care around the time of birth has been identified as a key strategy in most LMICs for reducing stillbirths, newborn and maternal morbidity and mortality. This is because studies have shown that the care a woman receives during pregnancy and childbirth has a huge and lasting impact on the woman's decision to seek health care in the future [4].

The WHO defines Respectful Maternal Care (RMC) as care that is organised for and provided to all women in a manner that ensures that their dignity, privacy and confidentiality is maintained. It also specifies guidance that ensures that women are free from harm and mistreatment in addition to being provided with informed choice and continuous support during labour and childbirth [5]. The Pan American Health Organisation (PAHO) Virtual Campus defines Respectful Newborn Care (RNC) as "an approach that focuses on the individual human rights which frame aspects related to ethics, rights and interpersonal relationships, including the respect for women and the newborn fundamental rights, such as autonomy, dignity, decision making and preferences" [6].

The issue of RMNC has received worldwide attention and several researchers have identified categories and typologies for RMNC as well as disrespect and abuse of women. Bowser and Hill [7], in their landscape analysis devised categories for disrespect and abuse of women which drew on human rights and ethical principles such as; physical abuse, non-consented care, non-confidential care, non-dignified care, discrimination based on specific patient attributes, abandonment of care, and detention in facilities. This typology however, had its limitations thus Bohren and colleagues conducted a mixed method systematic review to come up with an evidence-based typology which identified seven domains of mistreatment as follows; physical abuse, sexual abuse, verbal abuse, stigma and discrimination, failure to meet professional standards, poor rapport between women and providers and health system conditions and constraints [2].

These typologies have been a basis for the formulation of interventions that can be used to reduce and/or eliminate incidences of disrespect and abuse of women, thus promoting RMNC. The most common type of intervention for RMNC that is implemented especially in LMICs is training of health workers to be advocates for RMNC [8].

Core Outcome Sets (COS) are disease or healthcare specific important outcomes that can take into account both the potential benefits and harms of interventions [9]. The use

of COS reduces inconsistencies, allowing results from different studies to be compared and combined. They also help ensure that researchers are likely to report relevant outcomes and reporting bias may be minimised. This is because researchers are expected to report on all the core outcomes or state explicitly why particular outcomes are not reported [10].

The WHO acknowledges that coming up with and choosing the most important outcomes is critical to producing useful guidelines [11]. This study aims to develop a COS for RMNC which can be used by researchers in a LMIC setting.

## Purpose of the study

The purpose of the study is to develop a COS for RMNC that can be used in research studies and clinical practice in LMICs.

## Objectives of the study

1. To systematically review outcomes that are currently being reported in research studies on RMNC.

2. To use qualitative methods to establish the outcomes that are most relevant to key stakeholders of maternal and newborn care.

3. To involve key stakeholders in a consensus-based approach to agree on a COS that can be used in future research on RMNC as well as for routine care of women and newborns utilising maternity and newborn care services.

## Scope of the study

The study aims to develop COS for RMNC. The COS will be developed for research and clinical practice and will consider all interventions and care options for RMNC within this scope. This COS will predominantly be developed for use in an LMIC setting and will be for women who utilise maternal and newborn health services regardless of having had an institutional or home birth.

## Steering committee membership

The Study Steering Committee (SSC) will comprise membership from a multidisciplinary team within a NIHR Global Health Research Unit (GHRU) on the Prevention and Management of Stillbirths and Neonatal Deaths in Sub-Saharan Africa and South Asia. The team includes health care professionals, community engagement and involvement (CEI) representatives, including women who utilise maternal and newborn health services and methodologists (inclusive of a COS development expert) and experts from the United Kingdom with substantial collaborative research experience in an LMIC setting.

## Existing knowledge of outcomes

COS development projects relevant to the current study that have been published or ongoing were searched for in June 2024 in the COMET database. The studies were searched under disease category "Pregnancy and Childbirth" and disease name, 'maternal care and neonatal care'. There was no COS for respectful maternity care (RMC), respectful newborn care (RNC) or RMNC that was registered at the time of the search.

## Methods/design

The Core Outcome Set-STAndardised Protocol Items: the COS-STAP Statement [12] was followed for the development of this COS protocol. This study is registered on the Core Outcome Measures in Effectiveness Trials (COMET) database [13].

### Overview of the study design

This study will use the COS development approach guided by the COMET Handbook [14] and the Core Outcome Set-STAndards for Development (COS-STAD) recommendations [9]. An exploratory sequential mixed method design underpinned by the pragmatic philosophical perspective will be adopted for the study. This allows integration of both qualitative and quantitative research, thereby providing a better understanding of what the stakeholders especially the women and parents value and recommend to be included in the COS than either approach alone [15]. This design is also appropriate for the COS development approach which involves three steps shown in Fig 1.

### Step 1: Generation of initial list of outcomes

This step entails generating a list of outcomes by systematically reviewing existing literature from different databases on RMNC. These outcomes will be supplemented with those identified from secondary analysis of qualitative interviews of women who utilise maternal health services.

a)  Systematic literature review

**Research question.**  Which outcomes and outcome measures are currently being reported in trials and studies for RMNC research?

**Methods.**  A systematic literature review will be conducted in order to identify a list of outcomes currently being reported in studies on RMNC. The systematic review protocol for this study was prospectively registered on PROSPERO (Registration number CRD42023354521) [16]. The following databases will be searched from 2010 onwards: The Cumulative Index of

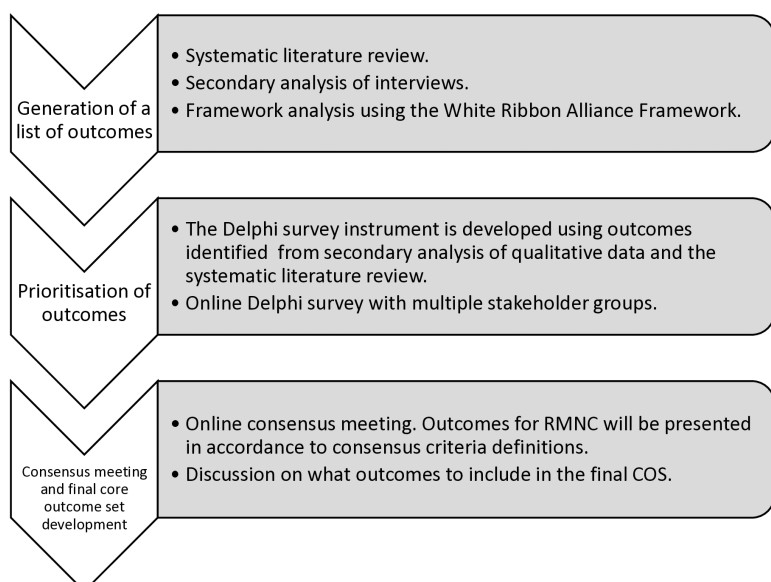

**Fig 1.  COS development approach for developing a COS for RMNC.**

Nursing and Allied Health Literature (CINAHL), Medical Literature Analysis and Retrieval System Online, (MEDLINE) and Global Health using the EBSCO host interface. The search will be limited to the year 2010 onwards as this coincides with the publication of the landscape analysis of Bowser and Hill [7] which was one of the first documents to bring the issue of disrespect and abuse of child labour to the forefront. Appropriate search strategies will be developed for each database using Subject Headings, MeSH Terms and Thesaurus and the following keywords: 'outcomes', 'respectful maternal and newborn care, respectful maternal care', 'respectful newborn care', 'disrespect', 'abuse', 'mistreatment'.

**Study eligibility.** In order to facilitate the identification of as many outcomes as possible, all types of studies that reported outcomes relating to RMNC will be included (for example, randomised controlled trials (RCTs), qualitative, quantitative, mixed methods studies and systematic reviews).

**Data extraction.** A data extraction form will be developed in advance to capture publication details of each included study in order to capture: the name of the author(s), year the article was published, name of the journal where the study was published, type of study, population, sample size, study site(s), the intervention(s) (where applicable), the outcomes that were reported and the tool used to measure the outcomes if appropriate. The articles will be screened and data extracted by FM and a 10% sample will be checked by two other authors (JJK and CB). Data extraction will be undertaken during the month of August 2024.

**Data analysis.** The methodological framework for summarising the outcome data will align to the White Ribbon Alliance (WRA) respectful maternity charter [17]. The charter contains ten rights (or domains) which focuses on high-quality respectful maternity care for both the woman and the newborn. These domains include but are not limited to; freedom from harm and ill-treatment, information and informed consent, dignity and respect, autonomy and self-determination, privacy and confidentiality, identity and nationality, highest attainable level of care, freedom from discrimination, non-separation of mother and baby, adequate nutrition and safe water. Extracted outcomes will be mapped to the most relevant section of the charter by FM with input from JJK and CB and presented in this way. As the purpose of the review is to identify only outcomes being reported for use in a COS development study, no evidence synthesis or quality assessment of the studies is required.

b) Secondary analysis of interviews

**Research question.** Which outcomes can be identified from the secondary analysis of interviews?

A secondary analysis of interview transcripts from Malawi, Tanzania and Zambia on RMNC will be undertaken. The NIHR programme of work includes further research projects related to RMNC in these countries [18–20]. Thus, the NIHR unit and the previous NIHR group has considerable data in the form of qualitative interviews conducted in these countries which can be accessed. The purpose of this secondary analysis is to capture outcomes suggested by parents (a key stakeholder) that may not have been captured in the literature identified by the systematic review. Interview transcripts (n = 33), will be analysed using the framework analysis method, providing a deductive and inductive approach [21]. Any new outcomes identified in this phase will then be classified according to the WRA framework discussed above.

**Review of final outcome list.** The final outcome list from Step 1) will be reviewed by members of the SSC to ensure that the outcomes have been mapped to the WRA framework accordingly, and outcome descriptions will be agreed upon with member of the CEI. The SSC will also take into account the number of outcomes and make suggestions to reduce or combine outcomes if not frequently mentioned. This step will ensure that the number of outcomes scored by the participants is manageable.

## Step 2: Prioritisation of outcomes (Delphi Process)

**Stakeholders.** Studies which have involved patient stakeholders and/or the public have come up with outcomes that may have not been previously identified by other stakeholders [10]. Once the outcomes have been identified and listed, stakeholders will be chosen from:

**Community Engagement and Involvement Group (CEI).** The CEI group will include women who are pregnant and those who have given birth within the past year. Their experiences will assist in refining the research question and identifying additional outcomes and prioritising them.

**Health care professionals.** Health care professionals involved in the care of women and newborns that is, nurses, nurse/midwives, obstetricians and paediatricians, will be invited to participate in the study.

**Researchers.** Researchers involved in different research areas of maternal and child health and core outcome set developers will be invited to participate in the study.

**Recruitment.** Participants representing the stakeholders identified above will be invited via email during the first two weeks of October 2024. The invitation will state the rationale for developing a COS for RMNC and how the Delphi survey will be administered. In order to increase the reach and sample size the participants may be asked to invite other professionals from their contacts who meet the criteria to also participate in the survey.

**Sampling and sample size.** A pragmatic approach will be used to select participants and determine the sample size to represent the health care workers, researchers and academics and women and their partners. Ideally, we will aim to recruit enough participants such that at least ten participants from each stakeholder group to complete both rounds of the Delphi survey in order to make reasonable inferences on the importance of each outcome.

Participants for the Delphi survey will predominantly be chosen from stakeholder groups identified above in Zimbabwe. Input may be sought from health care providers, researchers and academics from other LMICs in the Global Health Research Unit (GHRU), where additional ethical clearance is not required.

**Delphi survey.** A two-round online Delphi survey will be used to invite participants to score the outcomes identified in the systematic review and the secondary analysis of interviews. The Delphi approach enables participants' opinions to be sought and anonymously fed back for re-evaluation in sequential rounds [14]. The participants will be asked which stakeholder group they belong to (women and/or their partners, health care professionals and researchers). This will enable the collection of stakeholder specific data for the following groups:

Women and their partners:

- Age,

- Period since using maternity and/or neonatal services,

- Maternal and neonatal health services utilised (public government hospitals, public council clinics, private hospitals/clinics).

   Health workers:

- Profession (nurse/midwife, obstetrician, paediatrician etc)

- Country of practice

- Organisation affiliation (public government hospitals, public council clinics, private hospitals/clinics/practice).

   Researchers:

- Country of practice

- Area of research interest

- Organisation affiliation

**The Delphi rounds.** Participants will be asked to rate the importance of each outcome on a 9-point Likert scale (1-3 limited importance, 4-5 important but not critical and, 7-9 critical) following the GRADE (Grading of Recommendations, Assessment, Development and Evaluations) guidelines [22]. Participants will be given up to five days to respond after which a reminder is sent if there is no response. An assessment of the response rates for each of the stakeholder groups will be made after the first round. If there is a stakeholder group with a low response in the first round, reminders will be sent and if necessary further recruitment may be considered before commencing the second round.

Participants may also add any additional outcomes they think are important but not already included in the list. Any additional outcomes will be included in the second round of the Delphi survey following a discussion with the SSC about their inclusion. All outcomes scored in round 1 will be carried forward to round 2. Participants will be able to view the grouped stakeholder responses together with their own score in round 1 and will be asked to re-score the outcome based on this information using the same 1–9 scale. In round 2, participants may choose to change their score or to keep it the same.

**Consensus criteria.** On completion of the Delphi survey, the results will be summarised according to the pre-specified definition of consensus (Table 1).

## Step 3: Consensus meeting and final core outcome set development

**Consensus meeting.** It is now increasingly well accepted that the future of collaborative and influential research is to bring together diverse key stakeholders especially the patients to reach a consensus [10]. The consensus meeting will be done online. The results to all outcomes for RMNC will be presented in accordance to the consensus criteria definitions (Table 1) from the Delphi survey. Where outcomes from the Delphi reached 'consensus in' or 'consensus out', participants will be invited to briefly discuss and provide more information if they disagree with the inclusion/exclusion of the outcome in the COS. Following discussion, participants of the consensus meeting will re-score the outcome. Where outcomes were equivocal during the Delphi survey, they will be discussed and participants of the consensus meeting will be invited to re-score the outcome. Where voting is required, this will be undertaken anonymously using the same criteria and consensus definition as used in the Delphi survey (Table 1).

**Final core outcome set.** At the end of the consensus meeting, the consensus meeting panel will review the proposed outcomes to be included in the COS following the discussions and voting. A final reflective discussion will be undertaken to ensure the outcomes included are pragmatic and feasible to measure in an LMIC setting, and address any differences in

Table 1. Consensus criteria.

| Consensus Classification | Description | Definition |
|---|---|---|
| Consensus in | Consensus that the outcome should be included in the final core outcome set | 70% or more participants scoring as 7–9 (critical) AND < 15% participants scoring as 1–3 (limited importance) in all stakeholder groups |
| Consensus out | Consensus that the outcome should not be included in the final core outcome set | 50% or fewer participants scoring 7–9 (critical) in all stakeholder group |
| Equivocal | Uncertainty about the importance of the outcome | All other responses |

healthcare systems, socio-economic conditions and cultural factors. If a final COS is not agreed on at the end of the consensus meeting, subsequent online meetings will be considered in order to ratify the final COS.

## Ethical considerations

For the consensus part of the study (steps 2 and 3), ethical approval was sought from the Research Ethics Committee at Women's University in Africa and granted (Research Project No.: *1/03/2024).* Participant consent will be obtained in verbal (where possible) and then in written form. Ethical principles including voluntary participation, ensuring participant's right to privacy, anonymity and self-determination will be observed during the study and thereafter. Study participants will to the best of the researcher's ability be protected from any harm physically and psychologically.

## Discussion

Development of COS is essential in improving health care and reduce research waste. This will be the first COS to be developed for RMNC and will be predominantly used in the LMIC setting. It will be of particular importance in the LMICs where disrespect and abuse are more prevalent and where there is the highest burden of maternal mortalities [23].

There has been growing evidence from LMICs on disrespect and abuse of women during childbirth [2]. The researchers are confident that the research in Step 1 where the long list of outcomes will be generated will capture all outcomes relevant to all LMIC settings despite the social, cultural and economic differences. The inclusion of outcomes derived from secondary analysis of interviews will enable the capturing of outcomes from the patients' perspective. Until recently very few COS studies originated from LMIC and the development of COS rarely include a meaningful number of participants from the LMIC setting. This led to development of COS which include outcomes which may not be measurable in these settings [24].

This study protocol was also preprinted [25] and we would like to highlight two key changes from this original protocol which were decided upon before the commencement of this study. Firstly, there was an intention to include parents from six Sub-Saharan African countries and two countries from South Asia who are partners in the GHRU. Unfortunately, in our other recently completed COS study in stillbirth [26], obtaining ethical approval across eight individual countries where parent participants were involved took up to a year, which will be unfeasible for a PhD student project. Parent participation will therefore be limited to Zimbabwe from which the study is to be led. Secondly, there was an intention to use a real time Delphi process which has been used in recent COS development studies in the field of childbirth [27]. We obtained a year-long license to run the real-time Delphi for this study alongside the stillbirth COS development study but similarly due to delays in obtaining ethical approval, the license expired and thus we could not use the real-time Delphi software in the development of the RMNC COS. The decision to switch to a more traditional round-based Delphi approach was based on lack of funds to extend the real-time Delphi software license.

This study will be part of the few COS studies that have originated from LMICs and therefore it will help to understand the challenges that researchers who develop COS in LMICs face and how they can be mitigated. This information will help formulate and implement robust methodology in developing COS, ways to disseminate and implement in LMICs, thus ensuring the uptake and use of COS in these settings. As a starting point we will aim to disseminate this COS through our NIHR Global Health Research Unit (GHRU) Network and will liaise closely with our colleagues from the World Health Organisation with regards to further dissemination strategies.

## Supporting information

**S1 Checklist. COS STAP checklist RMNC.**
(DOCX)

## Acknowledgments

We would like to acknowledge the contributions from colleagues from the World Health Organisation, Portela A.G and Tuncalp O.

## Author contributions

**Conceptualization:** Farai Marenga, Jamie J. Kirkham.

**Funding acquisition:** Tina Lavender.

**Methodology:** Jamie J. Kirkham.

**Project administration:** Tina Lavender.

**Supervision:** Kushupika Dube, Unice Goshomi, Tina Lavender, Carol Bedwell, Jamie J. Kirkham.

**Writing – original draft:** Farai Marenga.

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
