## [Decision Letter · Decision Letter 0]

10 Dec 2024

PONE-D-24-27003Protocol for the Development of a Core Outcome Set for Respectful Maternal and Newborn Care in a Low-Middle Income Setting.PLOS ONE

Dear Dr. Kirkham,

Thank you for submitting your manuscript to PLOS ONE. After careful consideration, we feel that it has merit but does not fully meet PLOS ONE’s publication criteria as it currently stands. Therefore, we invite you to submit a revised version of the manuscript that addresses the points raised during the review process. Please submit your revised manuscript by Jan 24 2025 11:59PM. If you will need more time than this to complete your revisions, please reply to this message or contact the journal office at plosone@plos.org . Please include the following items when submitting your revised manuscript:

We look forward to receiving your revised manuscript.

Kind regards,

Zenewton André da Silva Gama, Ph.D.

Academic Editor

PLOS ONE

Journal Requirements:

“This research is funded by the National Institute for Health Care Research (NIHR) (NIHR 32027), a major funder of global health research and training, using UK aid from the UK Government to support global health research. The views expressed in this publication are those of the author(s) and not necessarily those of the NIHR or the UK Department of Health and Social Care.”

3. Please note that funding information should not appear in the Acknowledgments section or other areas of your manuscript. We will only publish funding information present in the Funding Statement section of the online submission form. Please remove any funding-related text from the manuscript. 

4. We noted in your submission details that a portion of your manuscript may have been presented or published elsewhere:

“A version of this protocol is preprinted which has been uploaded.”

**Additional Editor Comments:** <h4 style="font-weight: normal; line-height: 1.2; font-family: sans-serif, Arial, Verdana, "Trebuchet MS"; font-size: 13px; background-color: rgb(255, 255, 255); caret-color: rgb(0, 0, 0);">Strengths:</h4>

Relevance: The study addresses a critical issue—disrespect and abuse during childbirth in low- and middle-income countries (LMICs), which is key to reducing maternal and newborn mortality.Methodological Rigor: A mixed-methods approach, combining systematic reviews, qualitative analysis, and the Delphi process, ensures comprehensive and evidence-based outcome development.Stakeholder Involvement: Including a diverse range of stakeholders (women, healthcare professionals, and researchers) enhances the relevance and applicability of the outcomes.Ethical Considerations: Ethical rigor is maintained, with clear consent procedures and privacy protections.

<h4 style="font-weight: normal; line-height: 1.2; font-family: sans-serif, Arial, Verdana, "Trebuchet MS"; font-size: 13px; background-color: rgb(255, 255, 255); caret-color: rgb(0, 0, 0);">Recommendations:</h4>

Required Change – Clarify Use of LMIC Terminology:The broad use of LMICs may obscure significant socio-economic and healthcare disparities within this group. Action needed: Narrow the scope to low-income countries (LICs) or clearly discuss the impact of regional differences on the study’s outcomes.Recommended Change – Address Regional Variations:Differences in healthcare systems, socio-economic conditions, and cultural factors could influence the outcomes. Recommendation: Discuss these variations and how they may affect the outcomes identified in different LMIC settings.

<h4 style="font-weight: normal; line-height: 1.2; font-family: sans-serif, Arial, Verdana, "Trebuchet MS"; font-size: 13px; background-color: rgb(255, 255, 255); caret-color: rgb(0, 0, 0);">Conflicting Reviews:</h4>

Some reviewers may argue for the inclusion of all LMICs, while others may prefer a more specific focus on LICs. The authors should clarify whether the study will cover all LMICs or focus on LICs and explain the potential variations in outcomes across regions.

<h4 style="font-weight: normal; line-height: 1.2; font-family: sans-serif, Arial, Verdana, "Trebuchet MS"; font-size: 13px; background-color: rgb(255, 255, 255); caret-color: rgb(0, 0, 0);">Decision Justification:</h4>

Scientific Validity: The study is methodologically sound and transparent.Relevance and Impact: The study is relevant but may have limited impact if the outcomes are over-generalized across all LMICs without considering regional differences.

<h4 style="font-weight: normal; line-height: 1.2; font-family: sans-serif, Arial, Verdana, "Trebuchet MS"; font-size: 13px; background-color: rgb(255, 255, 255); caret-color: rgb(0, 0, 0);">Final Recommendation:</h4>

Required Change: Narrow focus to LICs or discuss the diversity within LMICs.Recommended Change: Address regional variations in maternal and newborn care across LMICs.

These changes will ensure the study aligns with PLOS ONE’s publication criteria.

Reviewers' comments:

Reviewer's Responses to Questions

**Comments to the Author**

1. Does the manuscript provide a valid rationale for the proposed study, with clearly identified and justified research questions?

Reviewer #1: Yes

2. Is the protocol technically sound and planned in a manner that will lead to a meaningful outcome and allow testing the stated hypotheses?

Reviewer #1: Yes

3. Is the methodology feasible and described in sufficient detail to allow the work to be replicable?

Reviewer #1: Yes

4. Have the authors described where all data underlying the findings will be made available when the study is complete?

Reviewer #1: Yes

5. Is the manuscript presented in an intelligible fashion and written in standard English?

Reviewer #1: Yes

6. Review Comments to the Author

You may also provide optional suggestions and comments to authors that they might find helpful in planning their study.

Reviewer #1: The study protocol presents an important justification for studies in the area of maternal and child health. Its methodology presents an aspect that should be highlighted: the inclusion of patients in the definition of outcomes. Because these patients belonged to low and medium development countries, the COS related to respectful maternal and newborn care will have a greater chance of being valid for this contexts.

The methodology is feasible and the description provide sufficient methodological detail for the protocol to be reproduced and replicated.

The authors justify the changes to the sample after the pre-print published.

The acronym for Study Steering Committee (SCC) in page 5 doesn't seem to be correct.

I have no suggestions. I believe it should be published.

7. PLOS authors have the option to publish the peer review history of their article (what does this mean? ). If published, this will include your full peer review and any attached files.

**Do you want your identity to be public for this peer review?** For information about this choice, including consent withdrawal, please see our Privacy Policy .

Reviewer #1: No

---

## [Author Response · Author response to Decision Letter 1]

17 Jan 2025

We address each of the editor and reviewer comments below in bold-italics and provide our responses in blue, indicating where we felt changes in the original manuscript was warranted.

https://journals.plos.org/plosone/s/file?id=wjVg/PLOSOne_formatting_sample_main_body.pdf [journals.plos.org] and

https://journals.plos.org/plosone/s/file?id=ba62/PLOSOne_formatting_sample_title_authors_affiliations.pdf [journals.plos.org]

>> We have reviewed and followed the template guidance.

“The views expressed in this publication are those of the author(s) and not necessarily those of the NIHR or the UK Department of Health and Social Care.”

>> The revised funding statement now reads – we will also include this in our cover letter.

“This research is funded by the National Institute for Health Care Research (NIHR) (NIHR 32027), a major funder of global health research and training, using UK aid from the UK Government to support global health research. The funders had no role in study design, data collection and analysis, decision to publish, or preparation of the manuscript."

3. Please note that funding information should not appear in the Acknowledgments section or other areas of your manuscript. We will only publish funding information present in the Funding Statement section of the online submission form. Please remove any funding-related text from the manuscript.

>> We have removed the funding statement from the manuscript.

4. We noted in your submission details that a portion of your manuscript may have been presented or published elsewhere:

“A version of this protocol is preprinted which has been uploaded.”

>> We are not really sure the context behind this query – a preprint is by definition not peer reviewed and is a posting for visibility rather than a publication. The route from MedRXiV (the preprint) to PLOS ONE is an allowable direct transfer in accordance with the guidance (Advancing the sharing of research results for the health sciences). We also mention this in the covering letter.

5. Please include captions for your Supporting Information files at the end of your manuscript, and update any in-text citations to match accordingly. Please see our Supporting Information guidelines for more information: http://journals.plos.org/plosone/s/supporting-information [journals.plos.org].

>> There are no supporting information files in this protocol.

>> No changes to references have been made and no references have been retracted.

Additional Editor Comments:

Recommendations:

1. Required Change – Clarify Use of LMIC Terminology:

The broad use of LMICs may obscure significant socio-economic and healthcare disparities within this group. Action needed: Narrow the scope to low-income countries (LICs) or clearly discuss the impact of regional differences on the study’s outcomes.

>> We will remain with LMIC as we can only involve participants from our partner countries. Our lead country for this project is Zimbabwe which is listed as LMIC (not LIC) according to the 2024/2025 World bank classification. World Bank country classifications by income level for 2024-2025. All our other participant countries also have LMIC status with the exception of Uganda.

Our search for potential outcomes to include in the COS comes from a variety of different research undertaken in different LMIC settings so we believe any societal or regional differences will be captured or come to light in the consensus building stages of this research. We have added a sentence on this in the protocol discussion.

2. Recommended Change – Address Regional Variations:

Differences in healthcare systems, socio-economic conditions, and cultural factors could influence the outcomes. Recommendation: Discuss these variations and how they may affect the outcomes identified in different LMIC settings.

>> Thanks for this comment. The final consensus meeting which will have representation from all countries will ask participants to comment on these factors once the final COS is agreed. We can then address these in our final reporting. We’ve extended our considerations to finalising the COS in the protocol to take into account these factors.

Reviewer #1: The study protocol presents an important justification for studies in the area of maternal and child health. Its methodology presents an aspect that should be highlighted: the inclusion of patients in the definition of outcomes. Because these patients belonged to low and medium development countries, the COS related to respectful maternal and newborn care will have a greater chance of being valid for this contexts.

The methodology is feasible and the description provide sufficient methodological detail for the protocol to be reproduced and replicated.

The authors justify the changes to the sample after the pre-print published.

The acronym for Study Steering Committee (SCC) in page 5 doesn't seem to be correct.

I have no suggestions. I believe it should be published.

>> Thank you for acknowledging the importance of our work and for recommending publication. We have changed the acronym SCC to SSC throughout.

---

## [Editor Report · Decision Letter 1]

2 Feb 2025

Protocol for the Development of a Core Outcome Set for Respectful Maternal and Newborn Care in a Low-Middle Income Setting.

PONE-D-24-27003R1

Dear Dr. Kirkham,

We’re pleased to inform you that your manuscript has been judged scientifically suitable for publication and will be formally accepted for publication once it meets all outstanding technical requirements.

Kind regards,

Zenewton André da Silva Gama, Ph.D.

Academic Editor

PLOS ONE

---

## [Editor Report · Acceptance letter]

PONE-D-24-27003R1

PLOS ONE

Dear Dr. Kirkham,

I'm pleased to inform you that your manuscript has been deemed suitable for publication in PLOS ONE. Congratulations! Your manuscript is now being handed over to our production team.

Kind regards,

on behalf of

Prof. Dr. Zenewton André da Silva Gama

Academic Editor

PLOS ONE